# Light therapy for sleep disturbance comorbid depression in relation to neural circuits and interactive hormones—A systematic review

**Chen Yaodong**[1]*, **Yingzi Zhang**[1], **Guo Feng**[2], **Yuanfang Lei**[1], **Qiuping Liu**[1], **Yang Liu**[1]

**1** School of Architecture, Southwest JiaoTong University, Chengdu, China, **2** Psychological Research and Counseling Center, Southwest Jiaotong Univerisity, Chengdu, China

* 971558522@qq.com

**Data Availability Statement:** All relevant data are within the paper and its Supporting information files.

## Abstract

### Aim

To provide an overview of the evidence on the effect of light therapy on sleep disturbance and depression, identify the light-active neural and hormonal correlates of the effect of light therapy on sleep disturbance comorbid depression (SDCD), and construct the mechanism by which light therapy alleviates SDCD.

### Methods

Articles published between 1981 and 2021 in English were accessed using Science Direct, Elsevier, and Google Scholar following a three-step searching process via evolved keywords. The evidence level, reliability, and credibility of the literature were evaluated using the *evidence pyramid* method, which considers the article type, impact factor, and journal citation report (JCR) partition.

### Results

A total of 372 articles were collected, of which 129 articles fit the inclusion criteria and 44% were at the top of the evidence pyramid hierarchy; 50% were in the first quarter of the JCR partitions. 114 articles provided specific neural and hormonal evidence of light therapy and were further divided into three groups: 37% were related to circadian regulation circuits, 27% were related to emotional regulation circuits, and 36% were related to hormones.

### Conclusions

First, neural and hormonal light-active pathways for alleviating sleep disturbance or depression were identified, based on which the neural correlates of SDCD were located. Second, the light responses and interactions of hormones were reviewed and summarized, which also provided a way to alleviate SDCD. Finally, the light-active LHb and SCN exert extensive regulation impacts on the circadian and emotional circuits and hormones, forming a dual-core system for alleviating SDCD.

**Funding:** National Natural Science Foundation of China (52008347, 32000734). The funders had no role in study design, data collection and analysis, decision to publish, or preparation of the manuscript.

**Competing interests:** The authors declare that they have no conflicts of interest to this work.

**Abbreviations:** ACC, anterior cingulate cortex; dlPFC, dorsolateral prefrontal cortex; DMH, dorsomedial nucleus of the hypothalamus; DRN, dorsal raphe nucleus; IGL, intergeniculate leaflet; LC, locus coeruleus; LGN, lateral geniculate nucleus; LH, lateral hypothalamus; LHb, lateral habenula; MHb, medial habenula; mPFC, medial prefrontal cortex; NAc, nucleus accumbens; PFC, prefrontal cortex; PVN, paraventricular nucleus of the hypothalamus; SC, superior colliculus; SCG, superior cervical ganglion; SCN, suprachiasmatic nucleus; SPVZ, subparaventricular zone; vLGN, ventral lateral geniculate nucleus; vlPFC, ventrolateral prefrontal cortex; VLPO, ventrolateral preoptic nucleus; VTA, ventral tegmental area.

# 1 Introduction

Numerous studies have reported that sleep and psychiatric disturbances (such as depression and anxiety) are comorbid and that each affects the other in a bi-directional manner [1,2]. For example, it was reported that 90% of patients with depression complained of sleep disturbance [2,3] and 80% complained of insomnia [4,5]; conversely, among patients with insomnia, the prevalence of depression was found to be 14–31% [1,6]. Other studies indicate that impaired sleep is a key symptom and risk factor for depression [6–9]. The intimate comorbidity of sleep disturbance and depression suggests the presence of overlapping processes, which may involve abnormalities in neurobiological and neuroendocrinological factors, among others. Although much progress has been made toward understanding how these disorders relate to each other, the biological basis of comorbidity remains largely unknown.

Light therapy is an evolving and non-intrusive treatment reported to be effective for many mental diseases related to neural disorders. Light is a critical in synchronizing circadian rhythms and bright light is known to regulate circadian rhythm and improve sleep disorders [10–12]. Regular exposure to daylight or to a specific intensity, wavelength, and duration of artificial light during daytime has been reported to be beneficial for treating mood and sleep disorders [11–16]. Some research has demonstrated that light therapy is more effective and faster-acting than the conventional antidepressant fluoxetine [17]. Conversely, light deprivation or aberrant light may induce depression-like behaviors and impair sleep [18].

Taken together, an overlapping pathology between sleep disturbance and depression comorbidities may exist. Light therapy on these comorbidities is reported to be beneficial at behavioral, neurological, and hormonal levels [11,13–15]. Therefore, it is reasonable to posit the presence of pathological sites at which light therapy might be therapeutic for these comorbidities. Therefore, identifying the pathological sites and contracting the mechanism of light therapy for alleviating SDCD are of great importance clinically and pharmacologically.

# 2 The review procedure

## 2.1 Aim

The aim of this study is to review the therapeutic evidence on the effect of light therapy on sleep disturbance and depression, identify the light-responsive neural and hormonal correlates of the effect of light therapy on SDCD, and construct the mechanism by which light therapy alleviates SDCD. Therefore, this review aims to achieve the following:

- Find therapeutic evidence on the effect of light therapy on sleep disturbance and identify the relevant light-active neural and hormonal pathways.

- Find neural and hormonal evidence regarding the effect of light therapy on depression and identify light-active neural and hormonal pathways.

- Identify the light-active neural and hormonal correlates of the effect of light therapy on SDCD.

## 2.2 Search methods for identification of studies

This study was conducted in accordance with the Preferred Reporting Items for Systematic Reviews and Meta-Analyses (PRISMA) statement [19] to ensure the quality of the review and the methodological considerations when using existing systematic reviews. We assessed risk of bias in the included articles using the revised Cochrane "risk of bias" tool. The disagreements

regarding bias and the interpretation of results were resolved through consensus reached by two researchers through discussion.

We searched the data using the Cochrane Methodology [20]. Journal articles were identified through computerized searches. PubMed and JSTOR were the databases used to collect journal articles, conference articles, books, and guidelines. This study mainly followed a three-step process to identify, collect, and screen the literature. According to the title and aim of this study, a group of preliminary keywords were identified, including light therapy, depression, sleep disorder/disturbance, circadian, and neural circuits.

For a further specific literature collection, some promising neural pathways, neural structures, and hormones were used in the PubMed and JSTOR research databases. The following combinations of keywords were selected: "light therapy" AND "depression"; "light therapy" AND "sleep disturbance"; "light therapy" [Mesh] AND "circadian rhythm"; "light therapy" [Mesh] AND "neural circuits"; "light therapy" [Mesh] AND "hormones"; "depression" AND "circadian rhythm"; "depression" AND "neural circuits"; "depression" AND "hormones"; "sleep disturbance" AND "circadian rhythm"; "sleep disturbance" AND "neural circuits"; "sleep disturbance" AND "hormones".

The screening process, shown in Fig 1, consisted of several different selection stages. After eliminating duplicate articles, the remaining articles were examined for further selection. In the next step, some promising neural pathways, neural structures, and hormones were identified, and a preliminary framework of the existing research was constructed. More specific keywords were then extracted, including combinations of "light therapy" with the names of related neural structures (e.g., suprachiasmatic nucleus/SCN, habenula, tegmental) or hormones (e.g., melatonin, serotonin, dopamine, glucocorticoids). In the final step, relevant literature identification. After a preliminary screening and selection of the literature, the names of the relevant neural structures and hormones were used as keywords to search evidence, useful conclusions, and studies within the full text of the selected literature.

## 2.3 Inclusion and exclusion criteria

The screening process was based on the following inclusion and exclusion criteria.

- Articles were limited to those published between 1981 and 2021 in English. The start year was chosen based on Lewy's benchmark publication in *Science* suggesting that light suppresses melatonin secretion in humans [10].

- Articles that were published in journals not cited in SCI were excluded; articles in the last quarter of JCR partitions gave priority to be excluded.

- The reliability and credibility of the articles that either fit the criteria or not were examined. The articles that were highly cited or evaluated to be reliable and credible were included, such as articles [21–28].

- Three researchers (YD-C, QP-L, and YF-L) independently reviewed titles and abstracts of the first 50 records and reached a consensus for all articles. Then, two researchers (QP-L and YF-L) independently screened titles and abstracts of all articles included. In case of disagreement, consensus on which articles to screen full-text was reached via discussion. If necessary, the third researcher was consulted to make the final decision. Next, two researchers (QP-L and YF-L) independently screened full-text articles for inclusion. Again, in case of disagreement, consensus was reached on inclusion or exclusion via discussion, and if necessary, the third researcher (YD-C) was consulted.

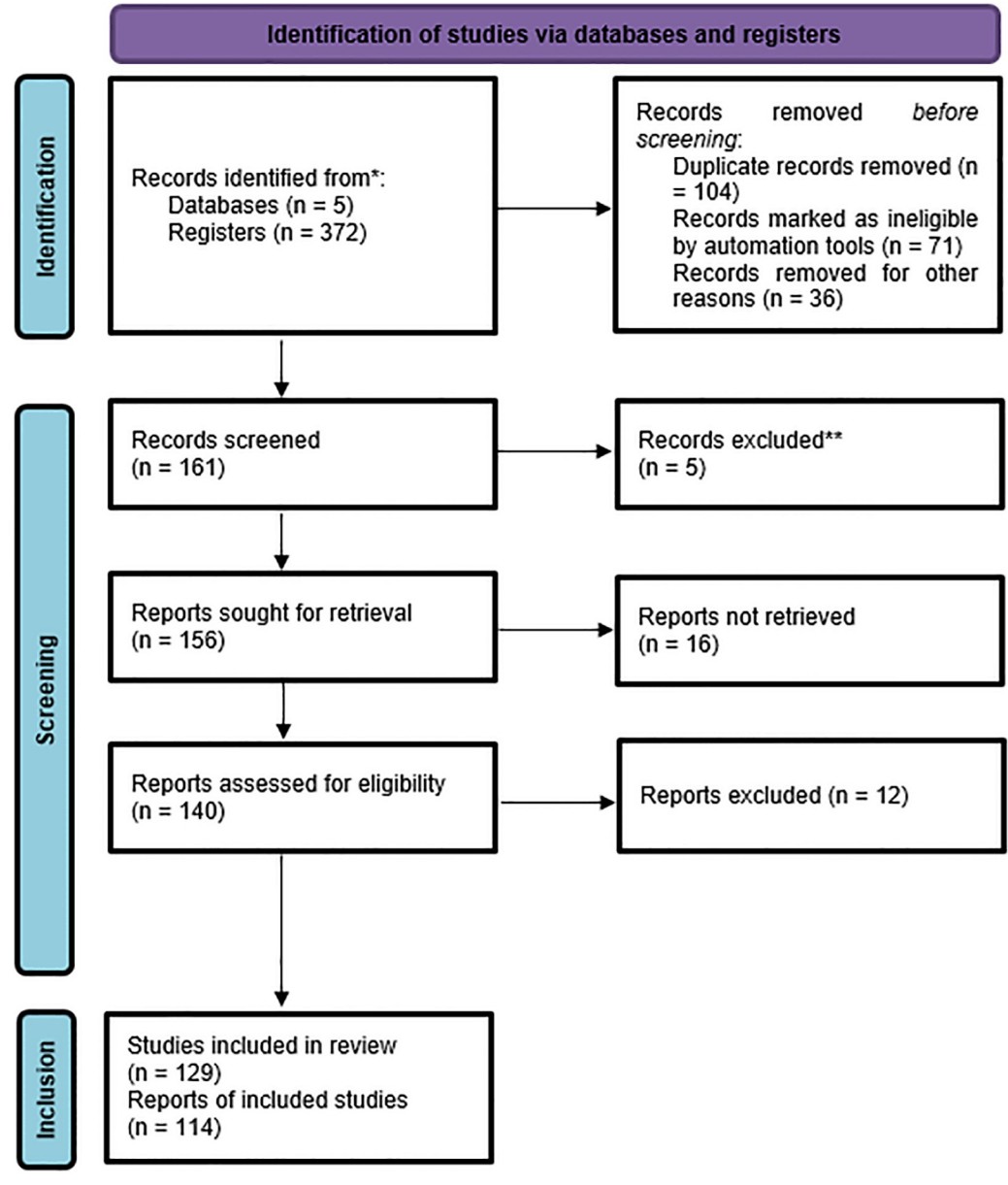

**Fig 1. PRISMA flow diagram.**

## 2.4 Analysis and classification

A total of 372 articles were identified and collected, 129 of which fit the inclusion criteria. After full-text reading and analysis, 114 articles provided specific neural and hormonal evidence of light therapy and were further divided into three groups (Fig 1): 37% were related to circadian regulation circuits, 27% were related to emotional regulation circuits, and 36% were related to hormones, as shown in Table 1. Furthermore, detailed information on each article, such as the impact factors, JCR partition, and article type, was collected. Through integrating with the evidence pyramid hierarchy suggested in the studies of Devlin et al. [29] and Samuel et al. [30], the evidence level of each study was evaluated according to article type, JCR

**Table 1. Classification and evidence levels of the included literature.**

| Topic and literature | | | | Keywords |
|---|---|---|---|---|
| Topic | Light therapy for sleep disturbance with comorbid depression | | | Light therapy, depression, sleep disorder/disturbance, circadian, neural circuits |
| Sub-topics | Circadian | SCN afferent and efferent pathways | [1]-3-CS, [2]-4-SR, [3]-3-CS, [4]-3-CCS, [5]-4-SR, [6]-4-RCT, [7]-3-CCS, [8]-3-CCS, [9]-3-CS, [10]-3-CS, [11]-4-SR, [12]-2- CR, [13]-2-CR, [14]-3-CCS, [15]-4-SR, [17]-4-RCT, [18]-4-RCT, [23]-4-SR, [24]-2-CR, [25]-2- CR, [26]-2- CR, [27]-3-CCS, [28]-3-CCS, [29]-3-CCS, [30]-4-RCT, [31]-3-CS, [32]-2- CR, [33]-4-RCT, [34]-3-CCS, [35]-4-RCT, [36]-4-SR, [37]-4-SR, [38]-3-CCS, [39]-3-RCT, [40]-4-SR, [41]-3-RCT, [42]-3-CCS, [43]-3-CCS, [44]-3-CCS, [45]-3-CCS, [46]-3-CCS, [47]-1-CR | Light, SCN, ipRGCs, retino-hypothalamic tract, PVN, pineal gland, sympathetic system, etc. |
| | | LHb afferent and efferent pathways | [13]-2- CR, [24]-2- CR, [25]-2- CR, [36]-4-SR, [48]-2- CR, [49]-2- CR, [50]-2- CR, [51]-2- CR, [52]-4-SR, [53]-2- CR, [54]-2- CR, [55]-4-SR, [56]-4-RCT, [57]-4-RCT, [58]-2- CR, [59]-4-SR, [128]-3-SR, [129]-4-SR | Light, LHb, VTA, DR, etc. |
| | Emotional | Cognition circuits | [60]-3-SR, [61]-4-SR, [62]-4-SR, [63]-4-SR, [64]-2- CR, [65]-3-CCS, [66]-4-SR, [67]-3-CCS, [68]-3-CS, [69]-3-CCS, [70]-4-RCT, [71]-4-SR, [72]-3-CS, [73]-3-CS, [74]-4-RCT, [75]-4-RCT, [76]-4-RCT | Light, emotional processing system, etc. |
| | | Adaptive stress response axis | [40]-4-SR, [46]-3-CCS, [77]-4-SR, [78]-2- CR, [79]-4-RCT | Light, adrenal cortex, etc. |
| | | Reward circuits | [55]-4-SR, [80]-3-CCS, [81]-3-CS, [82]-4-SR, [83]-4-SR, [84]-4-SR, [85]-3-CS, [86]-2-CS, [87]-4-SR | Light, DAergic activity, etc. |
| | Hormones | Hormonal interactions | [88]-4-SR, [89]-3-CS, [90]-3-CS, [91]-2- CR, [92]-4-SR, [93]-3-CS, [94]-3-CS, [95]-3-CS, [96]-3-CS, [97]-4-RCT, [98]-3-SR, [99]-3-CS, [100]-3-CCS, [101]-4-SR, [102]-2-CS, [103]-3-CCS, [104]-2- CR, [105]-4-SR, [106]-4-SR, [107]-3-CS, [108]-3-CS, [109]-3-CS, [110]-3-CS, [111]-4-SR, [112]-3-CS, [113]-2- CR, [114]-2- CR, [115]-4-SR | Light, melatonin, serotonin, dopamine, glucocorticoids, etc. |
| | | Light therapy and hormones | [116]-3-CS, [117]-3-CCS, [118]-3-CS, [119]-3-CCS, [120]-4-RCT, [121]-4-RCT, [122]-4-RCT, [123]-4-RCT, [124]-2-CS, [125]-4-RCT, [126]-4-RCT, [127]-4-RCT | |

Evidence levels: For Q1–Q3 articles, 1 = poor—expert opinion (EO), 2 = fair—case report (CR), 3 = good—cohort study (CS)/case control study (CCS), 4 = excellent—randomized controlled trial (RCT)/systematic review (SR); for Q4 articles, 0 = poor—expert opinion/case report (CR), 2 = fair—cohort study (CS)/case control study (CCS), 3 = good—randomized controlled trials (RCT)/systematic reviews (SR).

partition, and other factors, as shown in Table 1. For articles in the first three quarters (Q1, Q2, and Q3) of the JCR partitions, the evidence levels were 1 (poor) for the expert opinions, 2 (fair) for the case series and case reports, 3 (good) for the cohort studies and case control studies, and 4 (excellent) for the randomized controlled trials and systematic reviews. For articles in the last quarter (Q4) of the JCR partition, 1 was subtracted from the evidence level for each type of article. Fig 2 shows the distribution of the article types; 44% of the articles were randomized controlled trials or systematic reviews, which were at the top of the evidence pyramid hierarchy. Fig 3 shows JCR partition distributions; most of the articles were in the first three quarters of the JCR partitions, and 50% were in the first quarter.

## 3 Results

After screening, analyzing, and reviewing the selected literature, an overview of the evidence on the effect of light therapy on SDCD was obtained and presented in this section. Fig 4 illustrates the framework. First, evidence for the circadian and emotional regulation circuits and the light-response is presented, including i) light-responsive circadian circuits: the SCN afferent and efferent pathways, the LHb afferent and efferent pathways; and ii) light-responsive emotion pathways including cognition circuits, the adaptive stress response axis, and reward circuits. Second, the hormones and their light-triggered interactivities related to circadian and

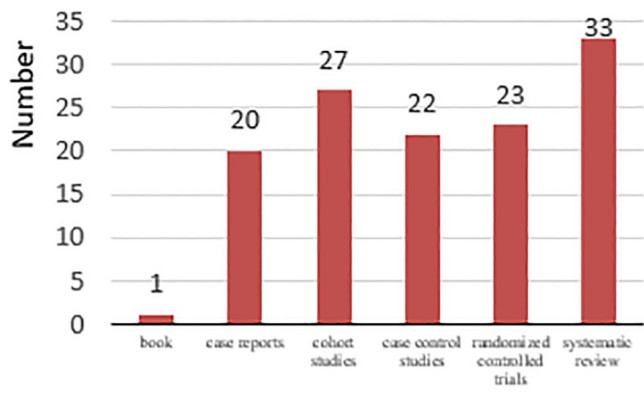

**Fig 2. Distribution of article types.**

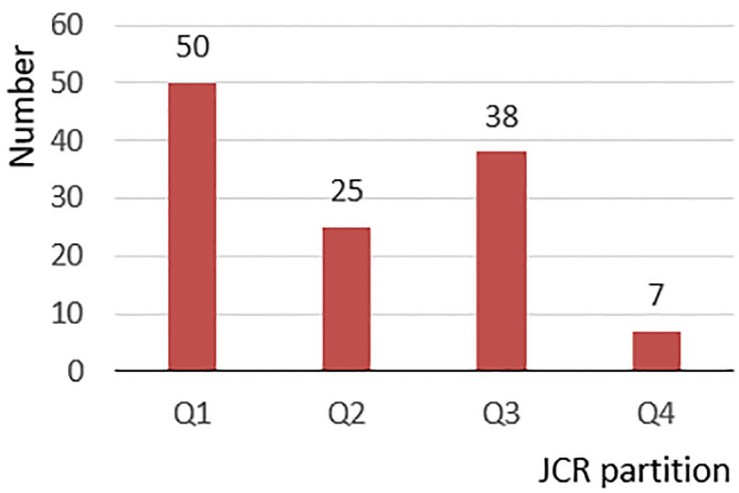

**Fig 3. JCR partition of the article types.**

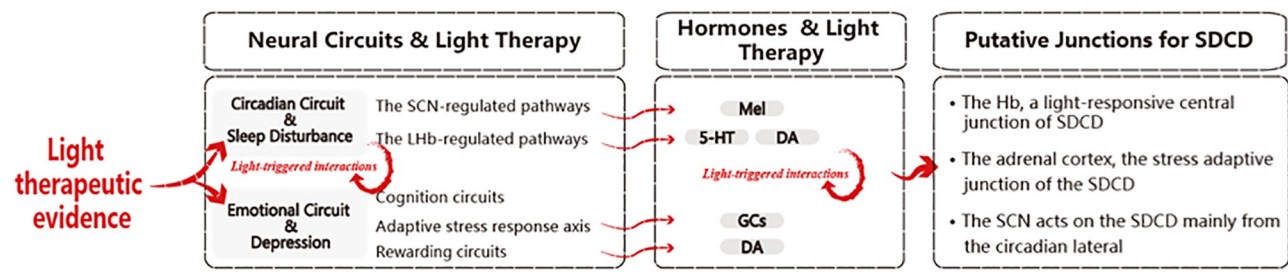

**Fig 4. The framework of this review.**

emotion regulation are presented. Third, taking all the evidence together, and emphasizing the interaction between the circadian and emotional neural and hormonal systems, we propose a mechanism of light therapy for these comorbidities, with several putative sites where light therapy may act to alleviate the pathology.

## 3.1 Circadian regulation circuits, sleep disturbance, and light

Sleep disturbances are behavioral disorders caused by disrupted circadian rhythms. A circadian rhythm is physiological and regulates behavioral cycles via a series of biochemical mechanisms headquartered in the hypothalamic suprachiasmatic nucleus (SCN, presents as an endogenous biological clock of human body), which has a periodicity of approximately 24 h. The rhythm is processed in the circadian regulation circuit, which includes multiple SCN afferent and efferent pathways. Any disruption in these pathways will therefore negatively affect numerous biological functions, among which, sleep disturbance is the most prevalent symptom.

Since 2002, the discovery of intrinsically photosensitive retinal ganglion cells (ipRGCs) [12] revealed that light plays an essential role in regulating circadian rhythm, and the mechanism of how light therapy regulates circadian rhythm has gradually been determined. The SCN synchronizes with the external photoperiod (and other time-givers) and regulates the rhythmicity of sleep–wake cycling and many other biological-clock processes [31]. It is via the ipRGCs that photoperiod signals provide input principally to the SCN, which are then relayed to relevant brain structures for performing circadian-related sub-functions through multiple neural pathways [32,33]. In recent years, increasing evidence has indicated that with the exception of the SCN, the lateral habenula (LHb) also contains an endogenous circadian oscillator and acts as another light-responsive central clock. Moreover, it interacts with the SCN as well [34–37]. Since then, a light-responsive dual-centered circadian regulation system has been recognized (Fig 5).

**3.1.1 The SCN afferent and efferent pathways.** IpRGCs→SCN is the classical afferent circadian regulation pathway suggested by Berson, Hattar, et al. [12,38]. The model of this pathway was modified after numerous subsequent studies. Light–dark signals reach the SCN mainly via the retino-hypothalamic tract (RHT) that connects the ipRGCs with the SCN [39–41] and also via an indirect projection from the IGL [42,43]. The SCN integrates and encodes the external and internal circadian-related stimuli into circadian signals (such as arginine vasopressin, or AVP, which is an active SCN-output neuropeptide that communicates with other brain structures and body organs for circadian regulation [44]) via a series of biochemical reactions prior to projecting them to the whole body via multiple neural pathways (Fig 5, yellow arrows).

The SCN-efferent melatonin regulation pathway is discussed first.

The SCN→PVN→SCG axis is the main efferent pathway through which the SCN regulates the pineal melatonin secretion for sleep-awake modulation [45]. Teclemariam-Mesbah et al. [46] injected pseudorabies virus (a powerful transneuronal tracer) into the pineal, which labeled the circadian regulation pathway, including the SCG, the intermediolateral column of the upper thoracic cord, the autonomic division of the PVN, and the SCN. It has been reported that systemic administration of AVP or electrical stimulation of the PVN both suppress the nocturnal melatonin surge [22]. In addition, the PVN can also receive external stress information from the brainstem and limbic forebrain area, encoded into CRH signals [47]. AVP and CRH were both reported to exert inhibitory effects on pineal melatonin [23,48]. Taken together, light therapy can regulate melatonin levels via the putative pathway: _light →ipRGCs →SCN (AVP↑) →PVN (AVP↑) →SCG (AVP↑) →pineal glands (melatonin↓)_. (Fig 5, yellow arrows).

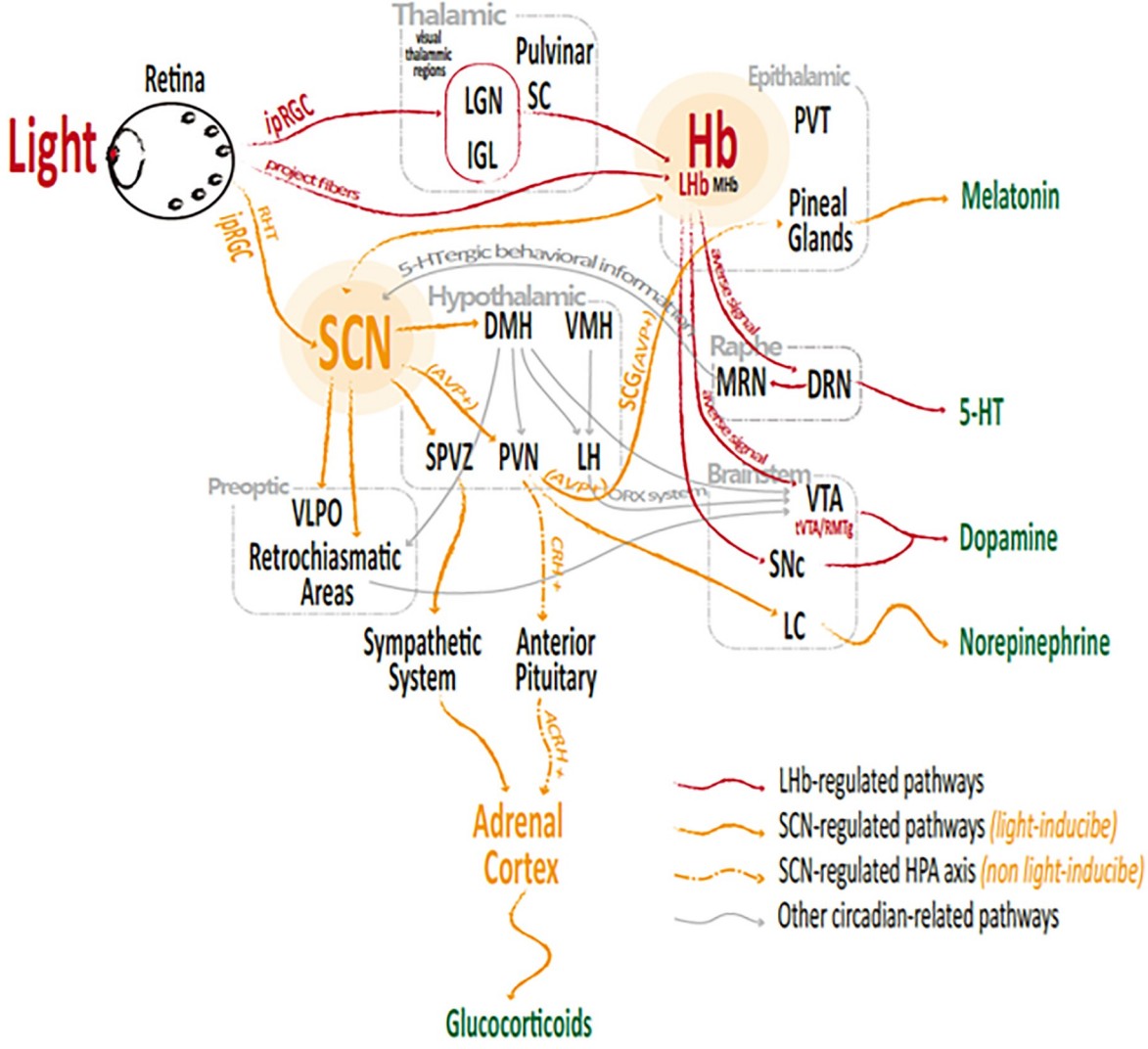

**Fig 5. Light-responsive pathways for circadian regulation.**

The SCN-efferent HPA regulation pathway is discussed second.

It has been reported that, at steady state (stress free), the hypothalamic-pituitary-adrenal (HPA) axis is the main efferent pathway via which the SCN regulates the circadian rhythmicity of glucocorticoids (GCs, which are a class of steroid hormones including cortisol and cortico-sterone) production and GC-related activities (such as arousal, immune, inflammation responses and sleep-awake behavior) [49,50]. Since the daily rhythmic, activation-inhibition cycle of the GCs participate in circadian regulation of peripheral organs via stimulatory and inhibitory effects on innate and adaptive immunity, this pathway coordinates the circadian oscillations between the central clock and peripheral organs. Balsalobre et al. [51] reported that the GC signal induces clock gene expression and resets circadian time in peripheral tis-sues. Further, Balbo et al. [49] reported that sleep onset is accompanied by an inhibitory effect on cortisol secretion, while awakening and sleep offset are accompanied by cortisol stimula-tion. Additionally, abrupt shifts in the sleep period trigger disruptions in the daily cortisol rhythm and sleep deprivation and/or reduced sleep quality seem to result in activation of the

HPA axis. However, light-induced adrenal activation has been reported not to be accompanied by ACRH variation [52], suggesting that light regulation on GC activity may not occur via the SCN→HPA axis. Taken together, the SCN regulates circadian rhythm of the peripheral organs via the putative pathway: _SCN (AVP) →PVN (CRH) →anterior pituitary (ACRH) →adrenal cortex (GCs)_ (Fig 5, yellow dotted arrows).

The SCN-efferent sympathetic-adrenal regulation pathway is discussed third.

As presented above, the adrenal secretory activity is reported to interact closely with sleep and emotion and the HPA axis is a primary component of the stress response. However, instead of the HPA axis, it has been found that light induces adrenal activation under SCN control, through the sympathetic-adrenal tract [52]. Ishida et al. [52] reported that light-induced nerve activation, gene expression, and corticosterone begin to increase 30 min after light stimulus, peaking at 60–120 min, and are evoked by the SCN-sympathetic-adrenal pathway. Niijima et al. [53] also reported a light-induced increase of sympathetic activity at 60–90 min, and this delayed activation is blocked by lesioning of the SCN, which along with the day-night difference of adrenal gene expression and corticosterone release, indicates that light-induced sympathetic-adrenal activity is closely linked to the circadian clock. The light-induced sympathetic-adrenal activation and plasma corticosterone surge indicate that light signals are instantly converted to glucocorticoid signals in the blood and cerebrospinal fluid. Thus, light induces adrenal secretory activities (such as corticosterone release) via the putative pathway: _light→ ipRGCs→ SCN (↑) → sympathetic system (SPVZ→ spinal cord→ IML↑) → adrenal cortex (GCs↑)_ (Fig 5, yellow arrows).

**3.1.2 The LHb afferent and efferent pathways.** IpRGCs project fibers directly to the peri-Hb region where the light–dark cycle modulates the activity of the LHb clock [13]. The ipRGCs also send light signals to the IGL and the LGN, which indirectly relay the signals to the LHb [13,32,33]. The LHb then projects circadian outputs (such as fasciculus retroflexus, fr [44]) to the various brain structures via multiple neural pathways (Fig 5, red arrows).

The LHb-efferent DA&5-HT regulation pathways.

The LHb (glutamatergic neurons) sends efferent circadian signals to the (GABAergic) rostromedial tegmental nucleus (RMTg, posterior to the VTA) and the raphe nuclei (RN), innervating the VTA DAergic and RN 5-HTergic activities for circadian regulation, respectively. Daily expression of clock genes (such as Clock, Per1 and Bmal1) has been reported in the VTA and SN, the two main sources of DA [54–56]. The release of these two hormones is reported to be both rhythmic and circadian. Converging studies have reported that the LHb encodes negative stimuli and that the LHb hyperactivity is inhibitory on VTA DAergic activity [57–59] and DRN 5-HTergic activity [60]. Lecca, Bourdy et al. [59,61] reported that the glutamatergic LHb cells project to GABA neurons in the RMTg, resulting in a reduction in firing of VTA DAergic neurons.

Light is reported to induce a decreased activation of the Hb, and this decrease is significantly greater in the morning than in the afternoon [62,63]. Consistent with this, light induces DA release, particularly the delivery of blue light pulses to the VTA [64]. Furthermore, a bright light therapy protocol has been found to attenuate negative signals, alleviate depression-like symptoms, and normalize excitability and burst firing in the LHb [58,65]. When chemogenetic manipulation was applied to the nodes of the ipRGC-vLGN/IGL-LHb circuit to block the pathway, the therapeutic effects were abolished. Taken together, light appears to regulate DAergic activity via the putative pathway: _light→ipRGCs→LHb (averse signal, glutamatergic neurons ↓) → RMTg (GABA neurons↓) →VTA (DA↑)_; and regulates 5-HTergic activity via putative pathway: _light→ipRGCs→LHb (averse signal, glutamatergic neurons ↓) →DR (5-HT↑)_ (Fig 5, red arrows).

## 3.2 Emotional regulation circuits, depression, and light

Our previous [66] study summarized the theory of the emotional processing system (EPS) [21], which suggests that neural activity in the cortex regulates the processing of cognition and emotion: Via bottom-up processes from the limbic structures (such as the amygdala, the HIP, and the ACC) to higher-level cognitive structures (such as the PFC, which is involved in modulating the emotional and cognitive response based on contextual information), emotional signals are detected, encoded, and assessed; via top-down processing, the activities of limbic structures are modulated, generating visceral feelings. These send outputs to regions involved in generating affective and expressive emotional responses [66], such as those of the reward system *(PFC-ACC/HIP/Amy- NAc-VTA)* that generate feelings of pleasure and to the HPA axis for the adaptive stress response. Increasing evidence suggests that functional failures in the EPS are key contributors to depression [66–70].

**3.2.1 Cognition circuits.**   A cluster of studies suggest that biased attention, biased processing, biased rumination, and biased memory for negative information are key elements consistently linked to the onset and deterioration of depression [66,67]. These processes involve the following circuits: biased attention, ACC—dlPFC- vlPFC -SPC- retina; biased processing, thalamus-Amy-ACC (sgACC, dorsal ACC) -PFC (dlPFC, vlPFC); biased rumination, Amy- HIP -sgACC-PFC (dlPFC, vlPFC, mPFC) and biased memory, Amy- HIP -caudate & putamen. It can be easily seen that emotional regulation involves thalamic, hypothalamic, limbic structures, and higher structures including the PFC. Within these circuits, in addition to the thalamic and hypothalamic regions, as presented in *section 1*, the limbic areas are frequently reported to be responsive almost immediately after light onset, due in part to their anatomical connectivity [66,71–79]. Bright light therapy is also reported to significantly increase Amy–prefrontal (mPFC) and intraprefrontal functional connectivity in a dose-dependent manner that is correlated with decreased anxiety symptoms [78,80]. Conversely, light dose suppresses threat-related prefrontal reactivity, with a higher bright-light dose associated with a lower mPFC response to fear and angry faces compared with neutral faces at rescan [80]. Furthermore, when synchronized with fear extinction and acquisition, blue light exposure instantaneously modulates prefrontal hemodynamic responses and alleviates fear expression after 24 h [81]. These data suggest that light can modulate top-down regulation of biased acquisition, processing, and rumination. Therefore, light may regulate cognition (attention, processing, rumination, memory) and emotion processing via *retina →limbic (Amy↔HIP /ACC) ↔PFC multisynaptic circuits* ([Fig 6](), purple arrows).

**3.2.2 Adaptive stress response axis.**   As presented in *section 3.1.1*, at steady state (stress free), the SCN regulates the HPA axis, whereas under stress, the circadian rhythm of the HPA axis and GC-related activity is disturbed, potentially leading to sleep disturbance and depression. Stress conditions (mainly from the brainstem and limbic forebrain areas [47]) are relayed principally to the PVN for adaptive response, following sensing, coding, and assessing of the EPS emotional significance. The PVN encodes the stress stimuli into CRH signals [47], which then induces the production of adrenocorticotropic hormone (ACTH) in the anterior pituitary. Finally, serum ACTH reaches the adrenal cortex and induces GC production [82]. Elevated GC levels negatively feedback on the PVN (CRH) and anterior pituitary (ACTH) [47], resulting in suppressed hippocampal neurogenesis [83]. However, excessive GC may damage neural structures such as hippocampal neurons, reducing hippocampal volume, a widely reported finding observed during depression [84]. Other studies [52] did not observe ACRH changes, suggesting that the light signal had not transcended the HPA axis, though apparently inducing adrenal secretory activation via the sympathetic-adrenal pathway. Taken together, the findings suggests that stress evokes an adaptive response in the peripheral organs via the

EPS→HPA axis pathway and activates adrenal secretory activity (*light → reward circuit → EPS → HPA axis → adrenal pathway*; Fig 6, yellow, purple and blue arrows). Secretory activity is also evoked via the *light → ipRGCs → SCN → sympathetic → adrenal pathway* (Fig 6, green arrows) and the *EPS*.

**3.2.3 Rewarding circuits.** In addition to the cognition and emotion processing circuits, reward circuits also play an important role in depression, by linking the circadian and emotional systems. It can activate DAergic activity centered in the VTA [61] and produce pleasure feelings from reward stimuli, which exert powerful and widespread positive responses over the whole body. Functional failures in the reward circuits may lead to anhedonia (loss of pleasure),

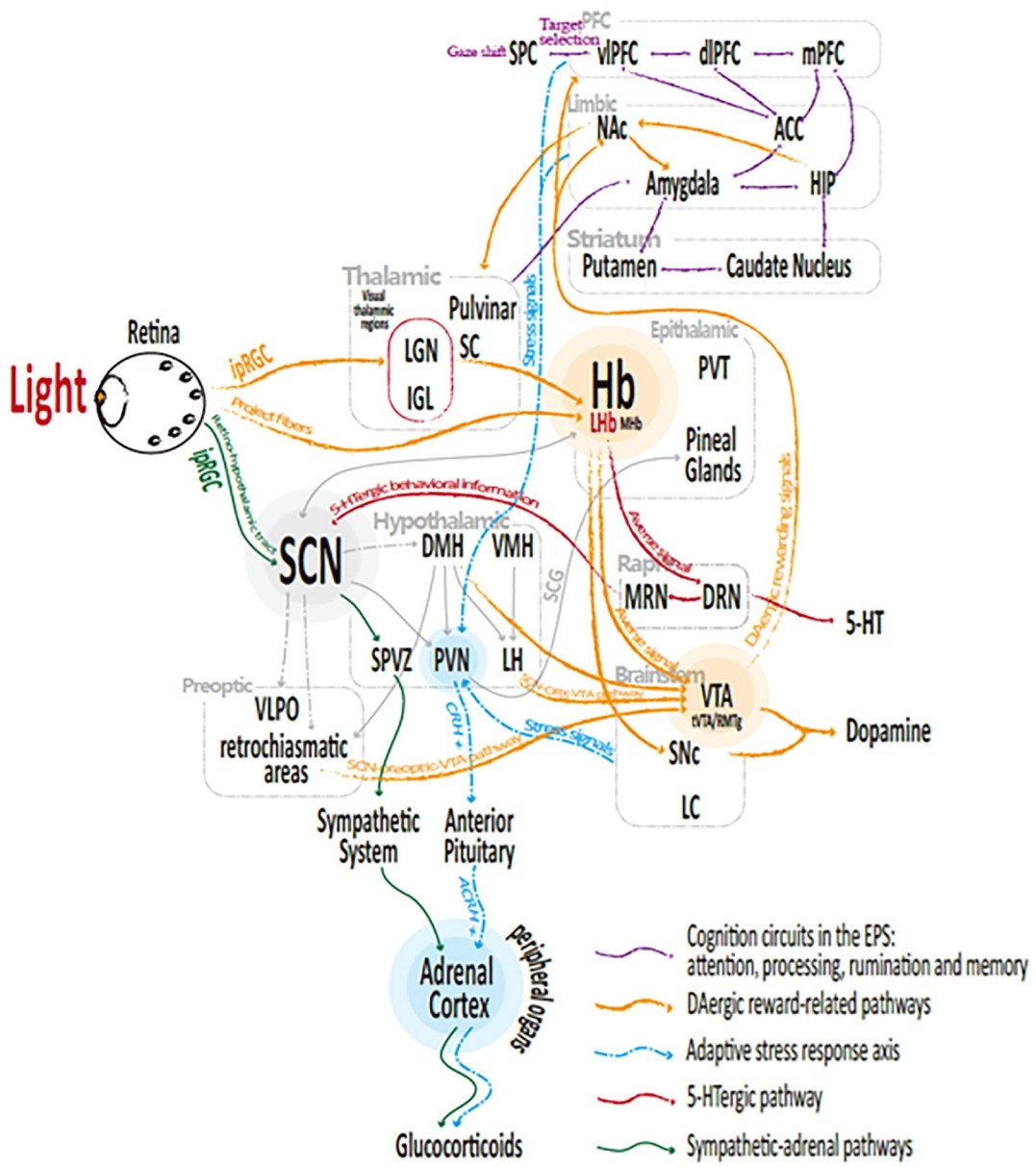

**Fig 6. Light-responsive pathways for emotional regulation.**

one of the most prevalent symptoms in depression [85]. The VTA regulates emotion via the DAergic reward loop [86,87], including a top-down circuit (ACC/PFC/HIP/Amy→NAc→VTA) that projects goal-directed behavioral information (mainly from the PFC) to the amygdala (involving in emotions and feelings) and Hip (involving in experience/memories) and finally to the NAc and VTA to activate the DAergic system [88]. Bottom-up circuits (VTA→NAc→ACC/PFC/HIP/amygdala), which release DAergic signals, relay back to higher structures, such as the amygdala, to modulate emotions and feelings, the PFC to modulate goal-directed behaviors and the Hip to modulate experience and memories [24,89,90]. Therefore, light may regulate emotion and circadian via the reward circuits (*light →ipRGCs→LHb→VTA ←→NAc ←→ACC/PFC/HIP/Amy*, Fig 6, yellow arrows). As the center of the reward circuits, the VTA is essential for emotional regulation, which, besides the LHb, can also be regulated by the SCN, via pathways such as the *SCN→preoptic area (PO) →VTA* and the *SCN→orexinergic system (ORX)→VTA* [91].

## 4 Interactive hormones and light therapy

The widespread hormones in cerebrospinal fluid and blood circulation are important elements via which the nervous system induces circadian and emotional responses. The functional failures in the nervous system can thus also be manifested in the hormonal activities. Among the hormones, melatonin (Mel), serotonin (5-HT, 5-hydroxytryptamine), dopamine (DA) and glucocorticoids (GCs) are the most important ones for circadian and emotional regulation. The hormonal properties are listed in Table 2, and the secretory rhythms are shown in Fig 7.

### 4.1 Hormonal interactions

The hormones exert antagonistic, stimulative, or inhibitory effects on each other, forming a complex and subtle system that keeps a rhythmic balance in circadian and emotion regulation. **First**, *DA & Mel*. Extant evidence suggests that a mutual inhibition may exist between melatonin and DA. Inhibition of dopamine release by melatonin was first observed by Zisapel et al. [105,106] in specific areas of the central nervous system (such as the hypothalamus and HIP); later, they elucidated the inhibitory mechanism of melatonin on dopamine release [107]. Findings in the retina [26,108] have also demonstrated that DA inhibits melatonin synthesis. **Second**, *5-HT & Mel*. Melatonin is metabolically derived from 5-HT [109]. Melatonin synthesis in mammals involves a four-step series of enzyme-catalyzed reactions [110] from Tryptophan to

**Table 2. Hormones and their properties.**

| Hormones | Secretion areas | Properties |
|---|---|---|
| Melatonin | Pineal glands | 1. Promotes sleep. [95] |
| | | 2. Direct marker of the circadian clock and sleep quality. [95–99] |
| Serotonin | RN, pineal glands | 1. Induces wakefulness by inhibiting REM sleep and the transition between non-REM sleep and REM sleep. [100] |
| | | 2. Mediates clock-resetting effects of behavioral stimulation. [101–103] |
| Dopamine | DAergic system, including the VTA and SN. | 1. Promotes wakefulness. [25,104] |
| | | 2. Involved in reward, motivational, and arousal activities. [25,104] |
| | | 3. Involved in the emotional adaptive response to stress and the immune system. [25] |
| Glucocorticoids | Adrenal cortex | 1. Involved in the emotional adaptive response to stress and the immune system; [25] |
| | | 2. Induces anti-inflammatory and immunosuppressive effects, resulting in bipolar effects on sleep and emotions. |

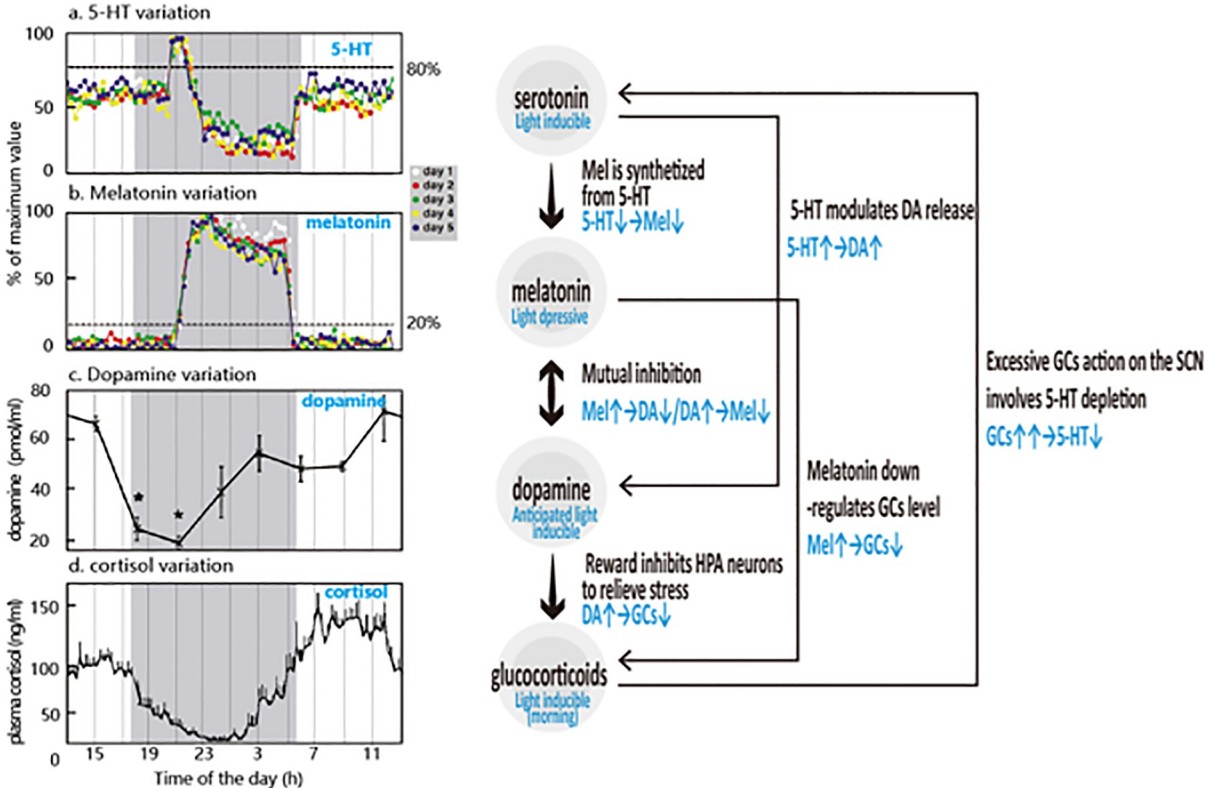

**Fig 7. Hormone daily variation (left) and hormonal interactions (right).** (Figs. a and b are re-drawn according to literature [92], Fig. c is re-drawn according to literature [93], and Fig. d is re-drawn according to literature [94]).

5-HT, 5-MT/ N-acetylserotonin (NAS), and finally to melatonin [111]. Liu et al. [92] reported the dynamic circadian rhythm of 5-HT, melatonin, and NAS in the pineal gland. They found that 5-HT-on preceded melatonin-on. A lag time of about 1 h exists for melatonin output after the 5-HT reaches its peak (then drops acutely, which may be due to the transformation into melatonin and NAS). On the whole, the findings indicate a close and direct interaction between 5-HT and melatonin. The lack of 5-HT at night may directly result in the suppression of nocturnal melatonin surges, leading to insomnia. **Third**, *5-HT & DA*. It is reported that 5-HT modulation of DA release in the basal ganglia is correlated with acute psychoses [112]. **Fourth**, *Mel & GCs*. Sainz et al. [113] found that melatonin decreases the levels of mRNA of the glucocorticoid receptor (GCR), suggesting that melatonin down-regulates GC levels. Quiros et al. [83] also reported melatonin regulates GC, but by reducing GCR nuclear translocation, instead of alterations in the mRNA levels of GCRs. Taken together, it is likely that melatonin can down regulate GC. Via this "melatonin↑→GCs↓" hormonal interaction, melatonin indirectly obtains immunomodulatory and anti-stress properties, can regulates antioxidant responses and apoptosis in immune cells as well [114]. **Fifth**, *GCs & Mel*. Szmyd et al. [115] concluded that GC action on the SCN involves both 5-HT depletion and reduced AVP signaling within the SCN, which may lead to elimination of the nocturnal melatonin surge, thereby resulting in sleep disturbances. Many facts support this conclusion and suggest that sleep disturbances are an underestimated steroid side effect [116,117]. For example, insomnia is listed as an adverse neurological reaction in FDA's (U.S. Food and Drug Administration) labels for

prednisolone and prednisone, alongside euphoria, mood swings, and depression [117]. Patel et al. [118] also suggested that insomnia was the most common adverse effect of steroids. These facts are consistent with the finding of Kellner et al. [23] that the CRH (which coordinates with GC activity within the HPA axis) has an inhibitory effect on pineal melatonin secretion in humans. **Finally**, _DA & GCs_. It has been found that reward inhibits paraventricular CRH neurons to relieve stress. PVN CRH neurons are rapidly inhibited by natural reward, and thus, reward relieves stress-induced behavioral and hormonal responses [119].

These findings are summarized and illustrated in Fig 7. The abnormalities in the hormonal activities underlie the onset and deterioration of SDCD. For example, stress may trigger insomnia and depression via the following hormone interactive circuit: _acute stress →GCs↑→5-HT↓→melatonin↓ (insomnia, which may turn into chronic stress) →GCs↑↑→brain damages (rewarding system damage → DA↓; hippocampal damages →biased cognition) →depression_.

## 4.2 Light therapy and hormones

The preceding hormones have been reported to be light-responsive, which is the basis for the application of light therapy in regulating circadian and emotional disorders. **For melatonin**, it is widely reported that light suppresses melatonin, varies its secretion rhythmicity in phase and amplitude, and does so in a dose-dependent manner [10,120]. **For 5-HT**, the current consensus is that of light induction. The monoamine hypothesis has been implicated with SAD for long time, is closely linked with the seasonal variation of light conditions. Therefore, 5-HT is believed to be associated with daylight. Lambert et al. [121] assessed the relation between 5-HT concentration and seasonal daylight conditions and found that the turnover of 5-HT was lowest in winter and that the rate of 5-HT production was directly related to the prevailing duration of bright sunlight, and rose rapidly with increased light intensity. Soler, Adidharma, Leach, et al. [79,122,123] exposed Nile grass rats (a diurnal rodent species) to daytime bright light (1000 lux) after dim light (50 lux) over a period of 4 weeks and found an increased number of DRN 5-HTergic and VTA DAergic neurons in animals that were housed in bright light (1000 lux). **For DA**, it has been reported that light stimulates DA release in the retina [124], as well as in the ventral striatum and NAc [67], and retinal DA synthesis and DA neuron activity display a dose-related interaction with white light [124]. Exposure to white light at 25 $\mu$W/cm$^2$ or greater appears to elicit the maximum response from retinal DA neurons. Brainard et al. [124] recommend a threshold irradiance around 3–5 $\mu$W/cm2. Since DA is released under the regulation of the reward system, it is subject to anticipation, which means that anticipated light may induce DA release, while unanticipated light suppresses it. For example, Romeo et al. [125] observed that after 90 days of continuous bright light exposure (which is surely not an anticipated light condition), DA neurons decreased progressively in the substantia nigra (SN) to 29%, paralleling a reduction of DA and its metabolite in the striatum. **For GCs**, GC activity is rhythmic and depends upon both the internal circadian clock and external dark-light cycles, and is associated with DAergic activity. Therefore, the influence of light on adrenal GCs remains controversial. However, in most studies, light exposure enhanced serum or salivary cortisol in the morning [94,126,127] but not in the evening. Scheer et al. [126] exposed 14 healthy men to darkness (0 lux) and bright light condition (800 lux, 1 h) on two subsequent mornings, and found that bright light exposure exerted a 35% increase in cortisol levels (besides the increase caused by the circadian rhythm) and cortisol levels 20 and 40 min after waking were significantly higher during exposure to 800 lux than during darkness. By contrast repeated experiments in the late evening demonstrated that light had no effect on cortisol levels. Thorn et al. [127] applied dawn simulation on 12 healthy participants, and observed a

significantly higher total production of cortisol in the participants that had applied dawn simulation during the first 45 mins after awakening. Touitou Y et al. [94] reported that a 2-h early awakening of bright light exposure increased the concentration of plasma cortisol and advanced its circadian phase. In addition, the effect of light on cortisol in the morning occurred in the absence of the sleep-wake transition [27]. Exposure to bright light in sleep-deprived participants (2000–4500 lux, 3 h) also induced an immediate elevation of cortisol levels in the early morning but not in the afternoon [128]. Leproult et al. [93] applied light exposure for 36 h transitioning from dim (<150 lux) to bright (4500 lux, 3 h) either from 05:00 to 08:00 am or from 13:00 to16:00 pm on subjects in constant wakefulness, and found that the early morning transition from dim to bright light induced an immediate, greater than 50% elevation of cortisol levels.

## 5 Summary and conclusion

In summary, in this study, we reviewed the therapeutic evidence on the effect of light therapy on sleep disturbance and depression and identified several neural and hormonal pathways that light therapy acts on to alleviate SDCD, including:

1. The SCN-efferent melatonin regulation pathway: light → ipRGCs → SCN (AVP↑) → PVN (AVP↑) → SCG (AVP↑) → pineal glands (melatonin↓) (Fig 5, yellow arrows).

2. The SCN-efferent sympathetic-adrenal regulation pathway: light → ipRGCs → SCN (↑) → sympathetic system (SPVZ → spinal cord → IML↑) → adrenal cortex (GCs↑) (Fig 5, yellow arrows).

3. The LHb-efferent DAergic /5-HT regulation pathway: light → ipRGCs → LHb (averse signal, glutamatergic neurons ↓) → RMTg (GABA neurons↓) → VTA (DA↑) (Fig 5, red arrows).

4. The LHb-efferent 5-HTergic regulation pathway: light → ipRGCs → LHb (averse signal, glutamatergic neurons ↓) → DR (5-HT↑) (Fig 5, red arrows).

5. The light-responsive reward circuit: light → ipRGCs → LHb → VTA ⟷ NAc ⟷ ACC/ PFC/HIP/amygdala (Fig 6, yellow arrows).

6. The light-responsive stress response pathway 2: light → ipRGCs → SCN → sympathetic → adrenal pathway (Fig 6, green arrows).

7. The light-responsive stress response pathway 1: light → reward circuit → EPS → HPA axis → adrenal pathway (Fig 6, yellow, purple and blue arrows).

8. The hormonal interaction pathway: acute stress → GCs↑ → 5-HT↓ → melatonin↓ (insomnia, which may turn into chronic stress) → GCs↑ → brain damage (reward system damage → DA↓; hippocampal damage → biased cognition) → depression.

Taking all the evidence together, light therapy is effective for alleviating sleep disturbance with comorbid depression (SDCD), and the light-responsive LHb and SCN form a dual-core system for alleviating SDCD.

*First, the LHb acts as a central junction of the SDCD.* As illustrated in Figs 5 and 6, extensive emotional and circadian pathways pass through the Hb (especially the LHb), anatomically and functionally; and both circadian and emotional properties were observed in the LHb. Therefore, there is a converging [13,28,65,129] consensuses that the LHb is a central junction linking emotional and circadian disorders and that hyper-activation of the LHb triggers sleep disturbances and depression. Bright light could normalize the hyper-activation and burst firing in

the LHb, attenuate the negative signals, and exert extensive LHb-related effects (including DAergic rewarding and 5-HTergic activities), which, consequently, could ameliorate the depression-like phenotypes and insomnia.

*Second*, *the SCN acts on the SDCD mainly from the circadian lateral.* The SCN is the core of the circadian system and the main circadian pacemaker. Since circadian and sleep disturbances are dependent risk factors for depression and melatonin is one of the most light-sensitive hormones, light therapy to regulate the SCN and melatonin-related circadian circuits is a classical and effective approach for alleviating sleep disturbances. Further, stabilized circadian and sleep rhythms exert positive drives for alleviating depression.

*Third*, *the adrenal cortex acts as a stress adaptive site of the SDCD.* The adrenal secretory activity exerts a powerful impact on sleep disturbances and depression and is both stress-responsive and light-responsive. Although via separate pathways, stress and light stimuli target the same termini (the adrenal cortex), and trigger GC activity. Even though the light-induced responses are still controversial, that the adrenal cortex and GCs can be modulated by light is unambiguous. Therefore, the adrenal cortex may also be an important target of light therapy for SDCD.

This systematic review collected relevant literature to provide an overview of the evidence on the effect of light therapy on SDCD. We approximately collected 120 articles, some of which were highly cited or evaluated as being reliable and credible. However, the limitation of this review was that we were unable to access the full text of some articles included, although we deemed them relevant after independently reviewing the titles and abstracts. Most importantly, we could not conduct meta-analyses of findings across reviews, due to the heterogeneity of the included systematic reviews and the relatively small proportion of studies reporting meta-analytic findings.

## Supporting information

**S1 Fig. HD images.** The high-definition version of the figures in this article.
(PDF)

**S1 Table. PRISMA 2020 for abstracts checklist.** Reporting systematic reviews in the abstract.
(DOCX)

**S2 Table. PRISMA 2020 checklist.** A guideline for reporting these systematic reviews.
(DOCX)

## Author Contributions

**Conceptualization:** Chen Yaodong.

**Data curation:** Chen Yaodong, Yuanfang Lei, Qiuping Liu.

**Formal analysis:** Chen Yaodong.

**Funding acquisition:** Chen Yaodong.

**Investigation:** Chen Yaodong, Qiuping Liu.

**Methodology:** Chen Yaodong.

**Project administration:** Chen Yaodong.

**Resources:** Chen Yaodong.

**Supervision:** Yingzi Zhang, Guo Feng.

**Validation:** Qiuping Liu.

**Visualization:** Yuanfang Lei, Qiuping Liu, Yang Liu.

**Writing – original draft:** Chen Yaodong, Guo Feng.

**Writing – review & editing:** Yingzi Zhang, Guo Feng.

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
