## [Decision Letter · Decision Letter 0]

28 Apr 2023

PONE-D-22-25633Light Therapy for Sleep Disturbance Comorbid Depression in Relation to Neural Circuits and Interactive HormonesPLOS ONE

Dear Dr. Yaodong,

Thank you for submitting your manuscript to PLOS ONE. After careful consideration, we feel that it has merit but does not fully meet PLOS ONE’s publication criteria as it currently stands. Therefore, we invite you to submit a revised version of the manuscript that addresses the points raised during the review process.

Your manuscript has been reviewed by a referee who is a recognized expert in this field. The reviewer recommended minor changes. Please amend your paper accordingly

We look forward to receiving your revised manuscript.

Kind regards,

Hubert Vaudry

Academic Editor

PLOS ONE

2. Please amend either the title on the online submission form (via Edit Submission) or the title in the manuscript so that they are identical.

“National Natural Science Foundation of China (52008347, 32000734)”

Reviewers' comments:

Reviewer's Responses to Questions

**Comments to the Author**

1. Is the manuscript technically sound, and do the data support the conclusions?

Reviewer #1: Yes

2. Has the statistical analysis been performed appropriately and rigorously? 

Reviewer #1: N/A

3. Have the authors made all data underlying the findings in their manuscript fully available?

Reviewer #1: Yes

4. Is the manuscript presented in an intelligible fashion and written in standard English?

Reviewer #1: Yes

5. Review Comments to the Author

Reviewer #1: Congratulations for the article, it was a pleasure to read it.

I suggest reviewing the sequence of bibliographic entries, because I don't understand the order starting from the third page. It would be appropriate to include a recent review on the subject DOI: 10.3233/JAD-200478

6. PLOS authors have the option to publish the peer review history of their article (what does this mean?). If published, this will include your full peer review and any attached files.

Reviewer #1: **Yes: **Daniela Smirni

---

## [Author Response · Author response to Decision Letter 0]

10 May 2023

Responses to Reviewers

Dear Reviewers,

The manuscript entitled “Light Therapy for Sleep Disturbance Comorbid Depression in Relation to Neural Circuits and Interactive Hormones—a Systematic Review" has been revised. In the decision email, we noticed several comments that require revisions and responses. The revisions and responses have been listed and categorized in this letter.

Best regards

journal requirements:

1. Please ensure that your manuscript meets PLOS ONE's style requirements, including those for file naming. The PLOS ONE style templates can be found at https://journals.plos.org/plosone/s/file?id=wjVg/PLOSOne_formatting_sample_main_body.pdf andhttps://journals.plos.org/plosone/s/file?id=ba62/PLOSOne_formatting_sample_title_authors_affiliations.pdf

##The manuscript is checked. 

2. Please amend either the title on the online submission form (via Edit Submission) or the title in the manuscript so that they are identical.

3. Thank you for stating the following financial disclosure: “National Natural Science Foundation of China (52008347, 32000734)” Please state what role the funders took in the study. If the funders had no role, please state: "The funders had no role in study design, data collection and analysis, decision to publish, or preparation of the manuscript." 

##The statement "The funders had no role in study design, data collection and analysis, decision to publish, or preparation of the manuscript." is added in the cover letter.

4. In your Data Availability statement, you have not specified where the minimal data set underlying the results described in your manuscript can be found. PLOS defines a study's minimal data set as the underlying data used to reach the conclusions drawn in the manuscript and any additional data required to replicate the reported study findings in their entirety. All PLOS journals require that the minimal data set be made fully available. For more information about our data policy, please see http://journals.plos.org/plosone/s/data-availability. Upon re-submitting your revised manuscript, please upload your study’s minimal underlying data set as either Supporting Information files or to a stable, public repository and include the relevant URLs, DOIs, or accession numbers within your revised cover letter. For a list of acceptable repositories, please see http://journals.plos.org/plosone/s/data-availability#loc-recommended-repositories. 

Any potentially identifying patient information must be fully anonymized. Important: If there are ethical or legal restrictions to sharing your data publicly, please explain these restrictions in detail. Please see our guidelines for more information on what we consider unacceptable restrictions to publicly sharing data: http://journals.plos.org/plosone/s/data-availability#loc-unacceptable-data-access-restrictions. Note that it is not acceptable for the authors to be the sole named individuals responsible for ensuring data access. We will update your Data Availability statement to reflect the information you provide in your cover letter.

##The data file is attached. 

##Reference list is checked.

Comments to the Author

1. Is the manuscript technically sound, and do the data support the conclusions?

Reviewer #1: Yes

2. Has the statistical analysis been performed appropriately and rigorously?

Reviewer #1: N/A

3. Have the authors made all data underlying the findings in their manuscript fully available?

The PLOS Data policy requires authors to make all data underlying the findings described in their manuscript fully available without restriction, with rare exception (please refer to the Data Availability Statement in the manuscript PDF file). The data should be provided as part of the manuscript or its supporting information, or deposited to a public repository. For example, in addition to summary statistics, the data points behind means, medians and variance measures should be available. If there are restrictions on publicly sharing data—e.g., participant privacy or use of data from a third party—those must be specified.

Reviewer #1: Yes

4. Is the manuscript presented in an intelligible fashion and written in standard English?

Reviewer #1: Yes

5. Review Comments to the Author

##No revision requirement is noticed in the above comments.

Reviewer #1: Congratulations for the article, it was a pleasure to read it.

I suggest reviewing the sequence of bibliographic entries, because I don't understand the order starting from the third page. It would be appropriate to include a recent review on the subject DOI: 10.3233/JAD-200478

##The sequence of the references is revised.

##The review on the subject DOI: 10.3233/JAD-200478 is cited in the paper.

6. PLOS authors have the option to publish the peer review history of their article (what does this mean?). If published, this will include your full peer review and any attached files.

Do you want your identity to be public for this peer review? For information about this choice, including consent withdrawal, please see our Privacy Policy.

Reviewer #1: Yes: Daniela Smirni

##No revision requirement is noticed.

---

## [Editor Report · Decision Letter 1]

19 May 2023

Light Therapy for Sleep Disturbance Comorbid Depression in Relation to Neural Circuits and Interactive Hormones—a Systematic Review

PONE-D-22-25633R1

Dear Dr. Yaodong,

We’re pleased to inform you that your manuscript has been judged scientifically suitable for publication and will be formally accepted for publication once it meets all outstanding technical requirements.

Kind regards,

Hubert Vaudry

Academic Editor

PLOS ONE
---

## [Editor Report · Acceptance letter]

23 May 2023

PONE-D-22-25633R1 

Light Therapy for Sleep Disturbance Comorbid Depression in Relation to Neural Circuits and Interactive Hormones—a Systematic Review 

Dear Dr. Yaodong:

I'm pleased to inform you that your manuscript has been deemed suitable for publication in PLOS ONE. Congratulations! Your manuscript is now with our production department. 

Kind regards, 

on behalf of

Dr. Hubert Vaudry 

Academic Editor

PLOS ONE